# Crosstalk between Epigenetics and Metabolic Reprogramming in Metabolic Dysfunction-Associated Steatotic Liver Disease-Induced Hepatocellular Carcinoma: A New Sight

**DOI:** 10.3390/metabo14060325

**Published:** 2024-06-08

**Authors:** Anqi Li, Rui Wang, Yuqiang Zhao, Peiran Zhao, Jing Yang

**Affiliations:** 1College of Basic Medical Science, Heilongjiang University of Chinese Medicine, Harbin 150040, China; anqili321@126.com (A.L.); zyq18833276938@163.com (Y.Z.); zhaopeiran1005@163.com (P.Z.); 2College of Pharmacy, Heilongjiang University of Chinese Medicine, Harbin 150040, China; wrdx@sina.com; 3Key Laboratory of Basic and Application Research of Beiyao, Heilongjiang University of Chinese Medicine, Ministry of Education, Harbin 150040, China

**Keywords:** hepatocellular carcinoma, non-alcoholic fatty liver disease, epigenetics, metabolic reprogramming, combinatorial therapy

## Abstract

Epigenetic and metabolic reprogramming alterations are two important features of tumors, and their reversible, spatial, and temporal regulation is a distinctive hallmark of carcinogenesis. Epigenetics, which focuses on gene regulatory mechanisms beyond the DNA sequence, is a new entry point for tumor therapy. Moreover, metabolic reprogramming drives hepatocellular carcinoma (HCC) initiation and progression, highlighting the significance of metabolism in this disease. Exploring the inter-regulatory relationship between tumor metabolic reprogramming and epigenetic modification has become one of the hot directions in current tumor metabolism research. As viral etiologies have given way to metabolic dysfunction-associated steatotic liver disease (MASLD)-induced HCC, it is urgent that complex molecular pathways linking them and hepatocarcinogenesis be explored. However, how aberrant crosstalk between epigenetic modifications and metabolic reprogramming affects MASLD-induced HCC lacks comprehensive understanding. A better understanding of their linkages is necessary and urgent to improve HCC treatment strategies. For this reason, this review examines the interwoven landscape of molecular carcinogenesis in the context of MASLD-induced HCC, focusing on mechanisms regulating aberrant epigenetic alterations and metabolic reprogramming in the development of MASLD-induced HCC and interactions between them while also updating the current advances in metabolism and epigenetic modification-based therapeutic drugs in HCC.

## 1. Introduction

Globally, HCC exhibits a 5-year survival rate of only 18%, emphasizing the poor prognosis of the disease [1]. With the increase in the prevalence of metabolic syndrome and antiviral drugs and alcohol control increasingly widespread, HCC from nonviral origins has been increasing rapidly, particularly in developed countries [2]. Non-alcoholic fatty liver disease (NAFLD), now better known as metabolic dysfunction-associated steatotic liver disease (MASLD), and its progression to non-alcoholic steatohepatitis (NASH), more recently referred to as metabolic dysfunction-associated steatohepatitis (MASH) [3], is becoming the most dominant risk factor for HCC [4]. Despite this, MASLD-related HCC has received relatively little attention due to MASLD patients being at a higher risk for cardiovascular events. The diagnosis of liver cancer is difficult in the early stages, and the prognosis is poor for patients who receive a late-stage diagnosis [5]. Understanding how MASLD progresses to malignant liver cancer lesions is key to early prevention and reversal of malignant transformation.

Epigenetic regulation is one of the hot topics in studying the pathogenesis and therapeutic intervention of HCC. Epigenetics research involves the changes in heritable gene expression or cellular expression that occur without altering DNA sequences [6]. Both the occurrence and development progress of MASLD and HCC are regulated by epigenetics [7], which is mostly reversible and high in plasticity [8]. It is, therefore, evident that epigenetic mechanisms play an important role in HCC initiation and progression, even in collaboration with other environmental influences.

Metabolic reprogramming refers to the process by which cells change their metabolic pathways and regulate related factors to adapt to specific environments or meet specific energy requirements [9], which has recently become a popular research area. As a result of the increased energetic and biosynthetic demands, cancer cells possess a distinct metabolic pattern that reduces oxidative stress and fulfills energy and biosynthesis needs [10]. Cancer cells have active pathways for nutrient uptake, synthesis, storage, conversion, and ATP production. These pathways include glycolysis, fatty acid (FA) synthesis, and amino acid metabolism [11]. Compared with virus-associated HCC, both MASLD and MASLD-induced HCC are associated with metabolic reprogramming. Consequently, linking the metabolic reprogramming that characterizes cancer with MASLD-dysregulated metabolism might be conducive to the discovery of novel metabolic readouts.

Epigenomes and tumor metabolism interact bidirectionally with the genetic and molecular factors that regulate cancer. On the one hand, alterations in the expression or activity of epigenetic enzymes can have a wide range of direct or indirect effects on cellular metabolism; on the other hand, metabolic reprogramming participates in epigenetic regulation by regulating epigenetic enzyme activity through fluctuations in metabolites and by transmitting information about the metabolic state of the cell to the nucleus [12]. A comprehensive understanding of key enzyme expression patterns and metabolic pathway interactions is crucial for appreciating the occurrence, progression, and treatment resistance of HCC. In certain contexts, metabolic alterations and epigenetic modifications may hinder immunosurveillance and facilitate immune escape, contributing to tumor progression. Since metabolic reprogramming in the immune cells is an important and very hot topic that has been reviewed elsewhere [13], it is only briefly mentioned in this review.

However, we still lack a comprehensive understanding of the tight interplay between epigenetic modifications, as well as metabolic reprogramming in MASLD-induced HCC and how their aberrant crosstalk affects tumorigenesis and progression. In this review, we focus on the mechanisms by which epigenetics and metabolic reprogramming influence and regulate the occurrence of HCC in the context of MASLD and update recent advances in the treatment of HCC by tumor-targeted epigenetic agents and employ targeting metabolic reprogramming. By gaining deeper insights into the interplay of these factors, we can develop more targeted and efficient therapies for HCC.

## 2. MASLD-Induced HCC

### 2.1. HCC Tumorigenesis

The intricate relationship between MASLD and HCC is influenced by the interplay of various pathogenic pathways. A theory known as the multi-parallel hit theory offers a more comprehensive explanation for the development of MASLD and its progression to HCC [14], including abnormal lipid metabolism, insulin resistance (IR), oxidative stress, inflammatory response, genetic alterations, and dysbiosis in gut microbiota (Figure 1). While most chronic liver diseases progress to cirrhosis before the onset of HCC, this sequence does not always apply to MASLD-related HCC. This is particularly true in cases of “lean MASLD”, where patients are without cirrhosis or obese. The development of HCC in these patients is believed to be influenced by factors such as endotoxin-related inflammation in the gut, as well as adipocytokines, leptin, and adiponectin [15].

Almost all the possible features of MASLD provide fertile ground for the advancement of hepatocarcinogenesis. This is largely driven by both lipotoxicity and IR, ultimately leading to increased fibrogenesis, inflammation, and abnormal cellular proliferation, as well as decreases in apoptotic cell death, necroptosis, and autophagy. CD36 is considered the primary driver of lipotoxicity associated with FAs in the progression of MASLD [16]. Recent research by Tao et al. demonstrated that the enhanced expression of CD36 could expedite the advancement of HCC by boosting the levels of aldo-keto reductases family 1 member C2 (AKR1C2) and enhancing the uptake of FAs [17].

The level of oxidative DNA damage in the liver is a significant factor in the development of HCC in patients with MASH. Compared with patients with HCC caused by other factors, those with MASH-induced HCC have been found to have significantly higher levels of oxidative DNA damage in their background liver [18]. Significantly, the development of oxidative stress and DNA damage in liver cells, specifically 8-hydroxy-2-deoxyguanosine (8-OHdG, a mutagenic DNA adduct resulting from lipid peroxidation), was identified as a critical factor in the onset of MASH-related HCC [19].

In the initial stages of tumor formation, autophagy inhibits tumorigenesis by eliminating aberrant cells. However, once a tumor is well-established, autophagy facilitates tumor growth and fuels HCC survival, thus contributing to the persistence and progression of cancer [20]. The impairment of autophagy in tumor cells due to metabolic stress leads to the elevated retainment of damaged mitochondria, enhancing oxidative stress and DNA damage [21]. Both the Western diet and fructose have been demonstrated to accelerate mitochondrial depolarization and mitophagy, resulting in mitochondrial dysfunction in the initial phases of MASH, which ultimately contributes to the progression of MASH toward HCC [22]. In MASLD/MASH, cardiolipin peroxidation was observed, and the oxidated cardiolipin represents a mitophagic signal to promote mitochondrial dismissal. Furthermore, cardiolipin frequently decreases during HCC progression, potentially serving as a mechanism to avoid apoptosis [23].

Familial, twin, and epidemiological studies have indicated that MASLD has a strong heritable component. Both common and rare mutations contribute to MASLD pathogenesis and the transition from MASH to HCC [24]. A recently identified MTTP rs745447480 variant, which encodes microsomal triglyceride transfer protein (MTP), causes progressive MASLD with subsequent cirrhosis and HCC in homozygotes [25]. Pinyol et al. identified the molecular specificity of NASH-HCC, with the TERT promoter, CTNNB1, TP53, and ACVR2A as the most commonly mutated genes. In particular, the mutation rate of the potential tumor suppressor gene ACVR2A is higher in NASH-HCC compared with other etiologies [26]. Remarkably, somatic mutations in MASH mice did not did not reveal increased tumorigenesis [27].

Changes in gut microbiota and metabolites in individuals with MASLD can result in the development of liver cancer [28]. The dysregulation of gut microbiota may cause abnormal epigenetic changes, impacting gene expression and playing a role in the pathogenesis of MASLD [29].

A substantial body of research indicates that the activation of the rat sarcoma virus (RAS)/rapidly accelerated fibrosarcoma (RAF)/mitogen-activated protein kinase (MEK)/extracellular signal-regulated kinase (ERK) pathway may contribute to HCC development [30]. The NASH microenvironment may damage hepatocytes, activating the RAS/RAF/MEK/ERK pathway. In this scenario, a hepatocyte exhibiting elevated MYC levels at the point of oncogenic transformation will give rise to an HCC [31]. RAS/RAF/MEK/ERK are the main signaling pathways associated with RAS. RAS family genes are the most prevalent proto-oncogenes in human tumors, and KRAS is the RAS member with the highest mutation rate and is strongly associated with poor prognosis in HCC [32]. For instance, in the CTNNB1 mutation, KRAS activates c-Met to promote HCC, and β-catenin inhibition effectively suppresses HCC progression [33].

MASLD-induced HCC development may also include the following mechanisms. For instance, recent progress in single-cell RNA sequencing has revealed hepatic stellate cells (HSCs) with unique functions [34]. HSCs can be categorized into myHSC and cyHSC subgroups, with the former promoting tumor growth and the latter inhibiting it [35]. Interestingly, not only the overall level of fibrosis in the liver but also the ratio of these different HSC subpopulations may contribute to creating a liver microenvironment that is more conducive to the development of tumors. The mechanisms of MASLD/MASH transition to HCC are briefly shown in Table 1.

### 2.2. Epigenetic Dysregulation in the Pathogenesis of MASLD-Induced HCC

Epigenetic alterations influenced by variables like age and environment may also influence the advancement of HCC related to MASLD. A growing body of proof has indicated that epigenetic modifications, including DNA methylation, histone modification, RNA modification, and non-coding RNA (ncRNA)-mediated processes, are significantly linked with the progression of MASLD and HCC [36,37].

#### 2.2.1. DNA Methylation

In genetics, methylation is a heritable enzyme-mediated chemical transformation catalyzed by DNA methyltransferases (DNMTs). This process involves transferring a methyl group from S-adenosyl methionine (SAM) to carbon 5 of the cytosine ring. DNA methylation primarily occurs at cytosine bases (C) when followed by guanine (G), thereby referred to as CpG sites. Generally, methylation in the promoter region results in transcriptional repression, whereas methylation in the gene region promotes gene expression—as described in the majority of cancers, including HCC [38]. In cancer, three types of aberrations in DNA methylation occur: hypermethylation of the CpGIs in the promoter regions of tumor suppressor genes, altered expression of DNMTs, and global hypomethylation of genes and repetitive sequences, thereby leading to genomic instability and oncogene activation [39].

Aberrant DNA methylation is a crucial initiating factor in the development of cancer in individuals with MASLD [40]. Tian et al. demonstrated the role of MASH-specific DNA methylation changes in the progression toward MASH-associated multistage HCC. This process induces gene silencing of genes implicated in DNA repair, lipid metabolism, glucose metabolism, and the progression of fibrosis via DNMT [41]. In particular, genes such as epidermal growth factor receptor, estrogen receptor 1, and glycine N-methyltransferase (GNMT) are methylated concurrently in cases of HCC and MASH with advanced fibrosis, supporting the idea of the simultaneous accumulation of CpG island methylator phenotype changes in the pathogenesis of HCC [42].

A growing body of research suggests significant and differential methylation of key genes responsible for lipid homeostasis, insulin signaling, DNA repair, liver tissue remodeling, and fibrosis progression during the progression of MASH to HCC [43]. In lean mice with MASH-HCC, variations in methylation were identified in genes associated with HCC progression and prognosis, as well as genes related to lipid metabolism. On the other hand, in obese mice with MASH-HCC, methylation differences were found in genes that were already known to be linked with HCC [44]. This highlights the importance of understanding the role of methylation in different populations with MASH-related HCC (Table 2), as it could provide valuable insights into potential treatment strategies and prognosis for patients.

#### 2.2.2. Histone Modification

Histone modifications play a key role in epigenetic processes that control transcription, DNA replication, and repair of damage, as well as the segregation of chromosomes during tumor development [45]. Histone proteins undergo various modifications, including acetylation, methylation, phosphorylation, ubiquitination, ribosylation, and SUMOylation. Among these, acetylation has been widely studied. Histone acetylase (HAT) promotes gene transcription by catalyzing histone acetylation, while histone deacetylase (HDAC) induces gene silencing through histone deacetylation. Herranz et al. showed that HAT1 induced in HCC plays important roles in promoting tumorigenesis and poor prognosis and is essential for the development of steatosis in mice [46]. Alongside this, HDAC8 has been identified as a key player in MASH-induced HCC, driving oncogenic pathways and chromatin modifications. Knocking down HDAC8 reverses IR and decreases tumorigenicity associated with MASLD [47]. Other research has demonstrated that the deacetylation of histone H4 lysine 16 leads to the suppression of cell death-related genes, contributing to the initiation of MASH-induced HCC [48]. In addition, FASN acetylation frequently decreases in human HCC and correlates with high levels of HDAC3. The use of HDAC3 inhibitors destabilizes FASN proteins and inhibits the growth of HCC (Table 2) [49].

**Table 2 metabolites-14-00325-t002:** Target genes related to DNA methylations and histone modifications in MASLD-induced HCC.

Mechanism	Experimental Model/Sample Data	Target Gene	Reference
DNA hypomethylation	NAFLD liver tissue and corresponding HCC tissue from HCC patients	DCAF4L2, CKLF, TRIM4, PRC1, UBE2C, TUBA1B	[41]
Gene promoter hypermethylation	Stelic mouse model of non-alcoholic steatohepatitis-derived HCC	GNMT, EGFR, ESR1	[42]
Differential methylation	Mouse models of lean and obese NASH-HCC	In lean NASH-HCC (CHCHD2, FSCN1, ZDHHC12, PNPLA6, LDLRAP1); In obese NASH-HCC (RNF217, GJA8, PTPRE, PSAPL1, LRRC8D)	[44]
Histone acetylation	The transcriptomic data of human liver samples were integrated from publicly available datasets	HAT1	[46]
Histone acetylation	LO2, HepG2, Bel-7404, and PLC5cells; Lentiviral-mediated shRNA knockdown in obesity-promoted NASH and HCC mouse models	SREBP-1	[47]
Histone acetylation	STAM NASH-related hepatocarcinogenesis mouse model	Cell death-related genes	[48]
Histone acetylation	HEK293T, HCT116, and ZR-75-30 cell lines; Human hepatocellular carcinoma samples	HDAC3, FASN	[49]

Abbreviations: DCAF4L2, DDB1 and CUL4 associated factor 4 like 2; CKLF, chemokine-like factor; TRIM4, tripartite motif-containing protein 4; PRC1, polycomb repressive complex 1; UBE2C, ubiquitin-conjugating enzyme E2C; TUBA1B; tubulin alpha 1b; GNMT, glycine N-methyltransferase; EGFR, epidermal growth factor receptor; ESR1, estrogen receptor 1; CHCHD2, coiled-coil-helix-coiled-coil-helix domain containing 2; FSCN1, fascin actin-bundling protein 1; PNPLA6, patatin-like phospholipase domain-containing protein 6; LDLRAP1, low-density lipoprotein receptor adapter protein 1; RNF217, Ring Finger Protein 217; GJA8, gap junction protein alpha 8; PTPRE, protein Tyrosine Phosphatase Receptor Type E; LRRC8D, leucine-rich repeat-containing 8; HAT1, histone Acetyltransferase 1; SREBP-1, sterol regulatory element-binding protein 1; HDAC3, histone deacetylase; FASN, fatty acid synthase.

#### 2.2.3. NcRNAs

Non-coding RNAs (ncRNAs) are a category of RNAs that lack protein-coding potency and can influentially alter diverse cellular processes and participate in pathogenic mechanisms. The class of ncRNAs in question includes microRNAs (miRNAs), long non-coding RNAs (lncRNAs), and circular RNAs (circRNAs). These molecules do not code for proteins but affect gene expression [50]. ncRNA may have significant regulatory functions in the initiation and progression of MASLD-HCC (Table 3).

Accumulating evidence suggests that miRNAs play a significant role in the epigenetic dysregulation of metabolic processes in MASLD and HCC. One of the most notable is miR-122. The reduction of miR-122 has been identified as a direct inducer of MASH-associated HCC [51]. Likewise, research conducted by Rodrigues et al. revealed that miR-21 acts as a crucial instigator of the whole-spectrum MASLD advancement and assumes an equally significant part in the evolution of MASLD to HCC [52]. It is worth mentioning that several miRNAs are differentially expressed in the transitions from MASLD to MASH to HCC, with some being overexpressed in tumors (miR-155, miR-193b, miR-182) while others are downregulated in HCC (miR-20a, miR-200c, miR-483) [53,54,55].

In the context of MASLD-HCC, lncRNAs are likely to impact the susceptibility to liver disease, potentially playing a role in the development of HCC [56]. HULC, the first lncRNA identified to be specifically overexpressed in HCC, has also been found to have increased expression in the liver tissue of MASLD rats [57,58]. Higher levels of lncRNA-PVT1 have been associated with advanced stages of MASLD in patients with HCC, indicating its potential as a diagnostic biomarker for identifying advanced MASLD stages [59]. Research has demonstrated that the activation of adipogenesis requires the lncRNA NEAT1 in a miR-140-dependent manner [60]. NEAT1 is associated with the development of liver fibrosis, MASLD, and HCC while serving as a preventative in the pathogenesis of liver failure by suppressing the inflammatory response [61]. Additionally, research by Wang et al. revealed that LINC01468 is upregulated in liver tissues during MASLD-HCC progression, and silencing this lncRNA can inhibit HCC tumorigenesis through lipid metabolism regulation [62]. These findings highlight the promise of lncRNAs as emerging prognostic markers and therapeutic targets in cancer treatment.

A multitude of conserved binding sites on circRNA function as a “miRNA sponge” by inhibiting miRNA activity through interactions with miRNA AGO proteins [63]. In HCC, certain circRNAs are dysregulated and can impact key processes associated with the progression from MASLD to HCC, including control over lipogenesis, fibrosis, and cellular proliferation [64]. While circMTO1 adversely affects the progression of HCC, CDR1 and circ_0067934 can enhance the invasion and metastasis in HCC [65]. The circRNA_0046366-related rebalancing of lipid homeostasis results in a significant reduction in TG, which in turn leads to amelioration of the hepatocellular steatosis phenotype. The upregulation of circRNA_0046366 abrogates the miR-34a-dependent inhibition of PPARα [66].

**Table 3 metabolites-14-00325-t003:** Relevant dysregulated ncRNAs associated with alterations in MASLD-induced HCC.

ncRNA	Experimental Model	Expression	Function	Reference
miR-122	NAFLD, NASH, HCC patients	Downregulated	Silence FRAT2 to avoid dysfunction of metabolism causing liver damage and the dysfunction of apoptosis through the dysregulation of TIMP1	[51]
miR-21	NAFLD-HCC patients, NAFLD-HCC mouse models	Upregulated	Through normalizing liver PPARα, miR-21 inhibition and suppression significantly reduced liver damage, inflammation, and fibrogenesis	[52]
miR-182	C57BL/6J mouse models were long-term HF or LF diet-fed	Upregulated	Shows early and significant dysregulation in the hepatocarcinogenesis process, and Cyld and Foxo1 as miR-182 target genes	[53,54]
miR-483	HCC patients, HepG2, SK-Hep1, and Hep3B cells, NAFLD mouse models	Downregulated	Inhibits cell steatosis and fibrogenic signaling	[55]
lncRNA HULC	Hep3B cells, HCC patients, NAFLD rat models	Upregulated	Promotes HCC growth and metastasis Promotes NAFLD development	[57,58]
lncRNA PVT1	NAFLD-HCC patients	Upregulated	Circulating could be a useful diagnostic biomarker for discriminating advanced stages	[59]
lncRNA NEAT1	HepG2, LO2 cells	Upregulated	Promotes adipogenesis, lipogenesis, and lipid absorption	[61]
LINC01468	THLE2 and the HCC cell lines, NAFLD-HCC patients	Upregulated	LINC01468-mediated lipogenesis promotes HCC progression through CUL4A-linked degradation of SHIP2	[62]
circMTO1	HCC patients, HCC-bearing male nude mice models	Downregulated	CircMTO1 inhibits HCC growth by upregulation of p21 via sponging miR-9	[65]
circRNA_0046366	HepG2 cells	Downregulated	Inhibits hepatic steatosis through miR-34a/PPARα	[66]

Abbreviations: TIMP1, tissue inhibitor of metalloproteinases-1; PPARα, peroxisome proliferator-activated receptor α; Cyld, cylindromatosis; Foxo1, forkhead box transcription factor O1; CUL4A, cullin 4A; SHIP2, inositol 5′-phosphatase 2.

#### 2.2.4. m6A Modification

Since the advent of high-throughput sequencing technology, over 170 distinct post-transcriptional RNA modifications have been noticed [67]. Most RNA modifications are found in transfer RNA and ribosomal RNA, but only a few have been found in mRNA. These include m6A, N1-methyladenosine (m1A), and 5-methylcytosine (m5C) [68]. Of the various modifications to mRNA, N6-methyladenosine (m6A) is by far the most abundant and well-studied. These modifications are dynamically reversible processes jointly regulated by three important types of proteins: writers (METTL3/METTL14/WTAP), erasers (FTO/ALKBH5), and readers (YTHDF1-3/YTHDC1-2) [69]. m6A is the most common and crucial alteration for controlling mRNA stability, splicing, and translation in MASLD and HCC [70]. m6A mRNA demethylation of IL-17RA has emerged as a key event before early tumor formation in HCC [71].

In terms of writers, Pan and colleagues discovered that the m6A methyltransferase METTL3 promotes MASLD-HCC [72]. METTL3 also enhances the expression and stability of LINC00958, which targets miR-3619-5p to upregulate hepatocellular carcinoma-derived growth factors, thereby promoting HCC lipogenesis and progression [73].

The m6A demethylase FTO has been widely studied due to its established relationship with obesity. Significantly increased mRNA and protein levels of FTO were observed in the livers of patients with MASLD, and large amounts of fat accumulated in the livers of the patients [74]. FTO levels are elevated in both HCC tissue and cells. Mechanically, this increase in FTO contributes to the growth of HCC by inducing demethylation of pyruvate kinase M2 (PKM2) mRNA and enhancing protein translation [75]. In contrast, Ma et al. found that FTO was downregulated in liver cancer tissues and inhibited the progression of HCC [76]. This is contrary to the fact that the demethylation regulation of FTO overregulates lipid metabolism in hepatocytes and promotes the development of HCC. Therefore, the m6A-modified “eraser” enzyme system FTO has different views on the regulation of HCC in different studies, and further research is needed.

In NAFLD livers, YTHDC2 inhibits the stability of mRNA for SCD1, FASN, SREBP-1c, and ACC1 and blocks their gene expression, resulting in the accumulation of TGs and the progression of MASLD [77]. YTHDF2, a predictor of poor HCC prognosis, promotes hepatocellular carcinoma stem cell phenotype and metastasis by upregulating octamer-binding transcription factor 4 [78]. However, it has also been demonstrated that YTHDF2 can act as a tumor suppressor. YTHDF2 binds directly to the RNA 3′-UTR of the epidermal growth factor receptor and promotes its degradation, thereby inhibiting HCC cell proliferation and growth [79]. In conclusion, epigenetic modifications may increase or inhibit the expression of specific genes, thereby affecting the progression of cells and maintaining cell stability in many ways. In the occurrence and progression of liver diseases, a variety of abnormal changes in epigenetic modification pathways are followed (Figure 2).

## 3. Metabolic Shifts and Reprogramming in MASLD-Induced HCC

MASLD-HCC grows slower than virus-induced HCC and acquires the ability to promote fat storage during carcinogenesis, i.e., “metabolic reprogramming” [80]. This section examines recent studies on the metabolic characteristics of HCC induced by MASLD, specifically looking at the changes in glucose, FA, and amino acid metabolism. These three key metabolic alterations have been a focal point in the study of HCC (Figure 3).

### 3.1. Glucose Metabolism

Glucose metabolism is vital for the development of MASLD-HCC [81]. Glucose participates in glycolysis, the pentose phosphate pathway (PPP), and the tricarboxylic acid (TCA) cycle, serving as the primary energy and nutrient supplier for cells.

#### 3.1.1. Glycolysis

The Warburg effect posits that cancer cells undergo a metabolic shift, prioritizing glycolysis over oxidative phosphorylation for energy production [82]. Despite glycolysis being less efficient than oxidative phosphorylation in terms of ATP production per glucose molecule, cancer cells benefit from the higher rate of ATP production through glycolysis (Figure 3A). This metabolic adaptation supports the rapid growth and proliferation of cancer cells [83]. Similarly, an enhanced glycolytic metabolic phenotype has been observed in the progression of HCC associated with MASH. The dysregulation of glycolysis processes plays a crucial role in driving the progression of MASLD to MASH, cirrhosis, and, ultimately, HCC [84]. This metabolic switching is prominent in patients with diabetic co-morbidity. The mutations stemming from MASH can facilitate IR and metabolic reprogramming, specifically an uptick in glycolysis even when oxygen is present, which may increase the risk of MASH-HCC [85]. Recent research has linked the presence of pathological AKR1B1 to metabolic reprogramming induced by hyperglycemia, which can lead to heightened lactate secretion and the Warburg effect in HCC. Thus, MASLD-associated HCC formation is promoted [86]. The altered activity of glycolytic enzymes is a frequent occurrence in HCC. For instance, the enzymes involved in the process of glycolysis (such as hexokinase 2, HK2, and PKM2) exhibit high levels of expression in HCC [87]. Additionally, the novel hexokinase called hexokinase domain containing 1 (HKDC1) has been identified as playing a role in liver cancer progression, with increased expression in MASLD and HCC [88]. In this study by Khan et al. [88], it was found that knockout of HKDC1 significantly impacts glucose flux, energy metabolism, and mitochondrial function, resulting in decreased ATP production. This, in turn, affects cell-cycle progression and induces ER stress.

KRAS mutations lead to the sustained activation of downstream signaling pathways, which, in turn, result in tumorigenesis, rewire cellular metabolism, and promote alterations in the tumor microenvironment [89]. Mutant KRAS is involved in glucose metabolism in multiple ways [90]. One such alteration is the Warburg effect [91]. The metabolic effects of oncogenic KRAS have been explained by transcriptional upregulation of glucose transporters and glycolytic enzymes [92]. A recent study has reported the direct role of the KRAS4A KRAS isoform in glucose metabolism through the regulation of the glycolytic enzyme HK1. This finding is significant because it further complicates the landscape of KRAS-mediated metabolism [93]. Moreover, the oncogenic KRAS has been shown to enhance mitophagy. The mutant form of KRAS activates a mitophagy receptor, NIX, which ultimately results in reduced mitochondrial function and increased glycolysis, thus promoting cell growth and enhancing redox balance [94]. 

#### 3.1.2. Pentose Phosphate Pathway

The PPP, a crucial branch of glycolysis, is responsible for producing ribonucleotides and providing NADPH [95]. HCC tissue demonstrates more active PPP when contrasted with neighboring normal tissues [96]. This heightened activity indicates the importance of PPP in meeting the needs of cancer cells for ribonucleotide synthesis and maintaining proper redox balance. As the rate-limiting enzyme in PPP, glucose-6-phosphate dehydrogenase (G6PD) has been the most studied one in HCC. G6PD has been identified as a key factor in promoting the migration and invasion of HCC cells in laboratory settings, primarily through facilitating the process of epithelial-to-mesenchymal transition (EMT) via the signal transducer and activator of the transcription 3 signaling pathway [43]. Metabolic analysis has revealed that the PPP plays a critical role in conferring resistance to regorafenib in HCC [97]. An increase in PPP function enabled cells resistant to regorafenib to better withstand the oxidative stress due to the medication. The interaction between G6PD, PI3K/AKT, NADK, and NADP+ might create a self-regulatory mechanism that controls the resistance to regorafenib in HCC. KRAS also stimulates the hexosamine biosynthesis pathway (HBP) and the non-oxidative arm of the PPP through MAPK-dependent signaling cascades, ultimately facilitating tumor survival [32].

#### 3.1.3. Tricarboxylic Acid Cycle

The TCA cycle is a critical metabolic pathway that controls cellular energy production and aids in the creation of macro-molecules while maintaining the cell’s redox balance. Studies have shown that the TCA cycle is upregulated during the progression of HCC. This change is believed to impact the function of enzymes involved in glutamine metabolism, malate/aspartate, and citrate/pyruvate shuttle, all of which are crucial for the development and advancement of HCC [98]. These disrupted metabolic pathways in the body are associated with impaired mitochondrial adaptation, leading to impairment in antioxidant activity and the disruption of ATP, which is critical in the transition from MASH to HCC [99]. Also, the production of reductive equivalents in the TCA cycle and their subsequent oxidation has a direct impact on the development of steatohepatitis through the induction of ER stress and reactive oxygen species production [100]. During glucose deprivation, cells activate p53 and peroxisome proliferator-activated receptor γ coactivator 1α (PGC1α) in a pathway that depends on AMP-activated protein kinase (AMPK). This activation of the AMPK-p38PGC1α axis ultimately benefits cancer cells by promoting oxidative metabolism.

### 3.2. Lipid Metabolism

A shared feature of HCC and MASLD is the reprogramming in lipid metabolism. The main characteristic feature of MASLD is the dysregulation of lipid metabolism, which is closely associated with the development of HCC [101] (Figure 3A). Our emphasis will be on how changes in lipid metabolism within cancer cells contribute to the progression of MASLD to HCC since the lipid metabolic rearrangement of immune cells within the tumor microenvironment (TME) has been reviewed in other studies [102].

#### 3.2.1. Fatty Acid Metabolism

The fundamental cause of MASLD is the abnormal accumulation of TGs due to disordered FA metabolism in liver cells. This indicates that the persistent dysregulation of FA metabolism may be the most basic mechanism underlying the progression from MASLD to HCC. There is currently a debate surrounding the mechanisms of FA uptake and utilization in HCC cells, and this continues to persist. One perspective argues that there has been a notable rise in the use of FA transport to meet the energy needs of HCC cells [17]. An alternative viewpoint suggests that HCC cells tend to engage in de novo lipogenesis (DNL) instead of depending on external sources of FAs [103]. This divergence in opinion could be attributed to differences in the functionality of various FA transporters.

In the context of HCC, lipoprotein lipase (LPL) has been demonstrated to boost tumor advancement by increasing the absorption of exogenous lipids. Specifically, both the LPL/FABP4/CPT1 axis and the Zinc-fingers and homeoboxes 2-LPL axis involved in FA metabolism reprogramming promote the transformation of MASLD to malignancy. Disruption of these pathways has been proven to halt HCC advancement effectively [104,105].

Normal cells mainly take up and acquire lipids through exogenous sources, whereas cancer cells are more dependent on DNL to maintain lipid homeostasis to satisfy their own proliferation and growth needs, and this process is accompanied by a high expression of sterol regulatory element-binding protein 1c (SREBP1c), fatty acid synthase (FASN), ATP-citrate lyase (ACLY), and stearoyl-CoA desaturase 1 (SCD1) [106,107]. DNL levels are elevated in patients with MASLD and HCC [108]. The accumulation of lipids in the liver leads to metabolic reprogramming, characterized by a confluence of cellular and metabolic modifications, alongside the accumulation of potentially harmful metabolites [109]. Serum metabolic profiling further revealed the alteration of metabolites associated with MASH-HCC. These altered metabolites are involved in the regulation of lipid metabolism through peroxisome proliferator-activated receptorα (PPARα), FA metabolism, biogenic amine synthesis, suppression of necroptosis, and the mechanistic target of rapamycin (mTOR) signaling pathway [110]. The oncogenic KRAS activates downstream signaling through the AKT, which eventually activates the ACLY enzyme, thus enhancing the conversion of citrate to acetyl-CoA and increasing DNL and sterol biosynthesis [111].

In HCC induced by MASLD, there is a tendency for the fatty acid oxidation (FAO) pathway to be suppressed, which serves to protect HCC cells from lipotoxicity [112]. MASH leads to an uneven distribution of oxygen within the liver lobules due to inflammation and the presence of fibrotic scars, ultimately resulting in a state of hypoxia. Patients with MASLD-induced HCC have been found to exhibit activation of the mTOR pathway, increased lipid accumulation, and upregulated hypoxia-inducible transcription factor (HIF)-2α [113]. Consequently, the activation of HIF-1α combined with the secretion of inflammatory cytokines could drive metabolic reprogramming, promote the growth of new blood vessels, and stimulate cell proliferation, ultimately facilitating the transition from MASH to HCC [114]. Fujinuma and colleagues demonstrated that the forkhead box K1 (FOXK1) protein inhibits the process of hepatic FAO in a manner dependent on mTORC1. Deletion of FOXK1 improves the progression from MASLD to MASH and, eventually, HCC by boosting the breakdown of lipids within liver cells [115]. ACSL3, a member of the long-chain acyl-CoA synthetase family, and ACSL4 play crucial roles in FA activation and are commonly upregulated in HCC [116]. In MASH, the overexpression of ACSL4 leads to enhanced steatosis by inhibiting FAO [117], whereas elevated ACSL4 levels in hepatomas contribute to heightened lipogenesis by indirectly boosting SREBP1 activity [118].

Lipid metabolism plays a crucial role in the communication between HCC cells and the TME. MASH contributes to the development of HCC by creating a proinflammatory microenvironment [119]. One critical aspect is the excessive activation of the classical inflammatory signaling pathway NF-kB. The activation of the NF-kB pathway can lead to the secretion of proinflammatory cytokines such as tumor necrosis factorα, interleukin-1 (IL-1), and IL-6, and transforming growth factor-b, which are linked to liver ailments [120]. These molecular signaling pathways promote the development of MASLD to HCC. Also, shifts in the immunological pattern compromised the hepatic immune system, transforming its anti-tumor function into a carcinogenesis-promoting process. Dysregulation of the gut microbiota through the gut-liver axis leads to immune activation and disrupted bile acid (BA) signaling, both of which have been shown to significantly impact the development and progression of MASLD [121].

#### 3.2.2. Cholesterol Metabolism

In addition to FA metabolism, cholesterol metabolic reprogramming in HCC cells can also promote the development of tumors [122]. Cancer cells maintain the production of cholesterol to support their growth and change the microenvironment, even in the presence of ample sterols. Research shows that disrupted cholesterol metabolism is a significant factor in the development of MASH and HCC [123]. Cholesterol synthesis was shown to promote the growth of HCC, even in the absence of FASN [124], which indicates a crosstalk between DNL and cholesterol synthesis. Liu et al. found that abnormally elevated cholesterol in HCC cells accelerated the process of malignant lesions from MASLD to HCC in mouse liver by inducing the lncRNA SNHG6 to localize in the ER-lysosome interaction region and activating the mTORC1 signaling pathway [125]. Importantly, positive correlations have been found between hypercholesterolemia and reductions in natural killer T (NKT) cells in patients with obesity, MASLD, and MASLD-related HCC. The study conducted by Tang et al. demonstrated that contrary to its role in triggering proinflammatory signaling in the liver, obesity-induced cholesterol accumulation through mTORC1/SREBP2 signaling activation specifically inhibits NKT cell-mediated anti-tumor surveillance within the liver [126]. Recent research by Li et al. demonstrated that signalosome 6 (CSN6) levels are elevated in HCC and act as an activator of hydroxymethylglutaryl-CoA synthase 1 (HMGCS1) in the mevalonate pathway, promoting the development of tumors [127]. The authors also illustrated that inhibiting CSN6 or HMGCS1 can potentially control the growth of liver cancer caused by MASLD. In addition, Hu et al. verified the idea that a feedback loop between cholesterol synthesis and the PPP contributes to the development of HCC. Inhibition of the PPP halted cholesterol formation, consequently hindering HCC in c-Myc mice [128].

It is becoming increasingly clear that the relationship between cholesterol metabolism, BAs, and HCC is significant, particularly in the context of MASH. The synthesis of BAs from cholesterol in hepatocytes acts not only as a digestive process but also as a signaling mechanism that influences the development of HCCV [83]. Trafficking of cholesterol to mitochondria through steroidogenic acute regulatory protein 1 (STARD1) is the rate-limiting step in the alternative pathway of BA generation [129]. Conde et al. discovered that STARD1 plays a significant role in MASH-driven HCC by promoting the generation of BAs in the mitochondrial acidic pathway. The products of this pathway have been found to stimulate hepatocytes, leading to pluripotency, self-renewal, and inflammation within the liver [130].

#### 3.2.3. MUFAs and PUFAs

During the progression of MASLD to HCC, there is a notable rearrangement of the serum lipidome. Research analyzing lipidomic profiles in human samples of MASLD-associated HCC has revealed a decrease in glycerophospholipids containing polyunsaturated fatty acids (PUFAs), like arachidonic acid (C20:4), accompanied by an increase in monounsaturated fatty acids (MUFAs), like oleic acid (C18:1) [131]. Nevertheless, it is unclear whether this dysregulation is a trigger for HCC or a compensatory mechanism that may rescue the phenotype. An increase in saturated fatty acids within cells and a decrease in membrane phospholipid unsaturation induced ER stress-associated cell death, and the activation of ER stress signaling plays an essential role in the onset of MASH-induced HCC. In a noncanonical pathway, MASH disrupts the negative feedback control of ring finger protein 43 (RNF43)/zinc and ring finger 3 (ZNFR3) in the WNT/β-catenin pathway [132]. Mutations in RNF43 and ZNRF3 alter lipid metabolism, specifically affecting unsaturated fatty acids and acyl-CoA biosynthesis in the context of MASH.

### 3.3. Amino Acid Metabolism

The liver is one of the central organs designed to control amino acid metabolism, which appears strongly enhanced in HCC patients [133]. In particular, HCC is usually accompanied by metabolic alterations in glutamine, branched-chain amino acid (BCAA), urea cycle, and one-carbon metabolism [134] (Figure 3A).

#### 3.3.1. Glutamine Metabolism

Rapidly proliferating cancer cells highly rely on glutamine for their energy needs [135]. The high demand for glutamine in rapidly dividing HCC cells results in a metabolic rewiring that leads to a “glutamine addiction” phenotype. This phenomenon is marked by HCC cells exhibiting an elevated glutamine uptake and subsequently increased glutaminolysis [136]. Normal liver cells mainly produce glutaminase2 (GLS2); however, in the progression of liver cancer, an alteration in metabolism driven by the MYC gene shifts the expression from GLS2 to GLS1, supporting the altered glutamine metabolism that occurs in HCC tumor cells [137]. Alternatively, high levels of GLS1 have been found to correlate positively with late-stage clinicopathological features in cancer and poor prognosis, possibly attributed to its capacity to trigger the pro-proliferative pathways Akt/GSK3β/cyclinD1 and ROS/Wnt/β-catenin [138,139]. Proline biosynthesis via glutamine may also be important for cancer cell growth [140]. Tang et al. demonstrated that the metabolic axis involving glutamine, proline, and hydroxyproline exerts its effects by modulating the activity of the HIF1a in HCC and contributing to resistance to the drug sorafenib [141]. More importantly, these findings indicate that a hypoxic TME exists in HCC. In HCC, the conversion of glutamine into α-ketoglutarate is utilized to sustain glucometabolic intermediates in a glucose-deprived TME [142]. This metabolic reprogramming, particularly evident during metastasis, allows for the continued operation of the TCA cycle through anaplerosis in nutrient-deficient HCC cells. This metabolic adaptability provides a survival advantage to HCC cells [143].

#### 3.3.2. Branched-Chain Amino Acid

BCAAs are a group of necessary amino acids comprising leucine, isoleucine, and valine. According to Takegoshi and colleagues, incorporating BCAAs into the diet of mice can halt the advancement of MASH and deter the emergence of HCC [144]. By contrast, the impairment of BCAA catabolism has been linked to the development and progression of HCC, as well as a decrease in overall survival rates [145]. Tajiri et al. found that levels of leucine were lower in MASH-HCC than in MASH patients. Pathway analysis revealed a notable increase in leucine and isoleucine degradation pathways [146]. In a recent study, Ahmed et al. also observed a reduction in the amino acid leucine in MASH-HCC patients compared with those with MASH [110].

#### 3.3.3. Urea Cycle

The urea cycle, also known as the ornithine cycle, is vital for preventing the buildup of ammonia levels [138]. The ammonia buildup in the body due to reduced urea cycle function could be a considerable mechanism for MASLD to deteriorate into HCC [147]. The urea cycle dysfunction in MASLD is thought to be linked to changes in the regulation of genes responsible for urea cycle enzymes and an elevation in liver cell senescence [148]. The overabundance of fructose acquired through the Western diet causes uric acid production, generating reactive oxygen species (ROS) in both liver cells and fat cells. The persistent suppression of the urea cycle in liver cancer changes the metabolic pathway from producing arginine to synthesizing pyrimidines [85]. In patients with HCC, there is a notable decrease in the expression of genes related to key enzymes of the urea cycle and associated metabolites, such as citrulline, arginine, and ornithine [149].

#### 3.3.4. One-Carbon Metabolism

One-carbon metabolism provides a substrate for methylation and is also a crucial pathway for the formation of nucleotides and reductants. The metabolism of serine, glycine, and methionine is closely interconnected with the production of 1C units. Overall, one-carbon metabolism is a complex network of interconnected pathways. A recent study by Li et al. also showed that dietary folate supplementation in DEN/HFD-induced mouse models promoted tumor development due to folate interactions with methionine and 1C metabolism in the liver [150]. 1C units derived from glycine in HCC cells support the progression of tumors by promoting purine and pyrimidine biosynthesis through the flux of the glycine cleavage system [151]. The downregulation of GNMT and betaine homocysteine methyltransferase, two crucial enzymes in the methionine cycle, has been documented in HCC. The regulation of these enzymes is disrupted in the context of HCC, potentially impacting the one-carbon metabolism [152]. In addition, not only are many enzymes involved in one-carbon metabolism altered in HCC, but they are also affected in the setting of MASH [153].

## 4. Interactions between Epigenetic Modifications and HCC Cell Metabolism

A bidirectional regulatory mechanism between metabolic remodeling and epigenetics exists in tumors, where many intermediate metabolites can act as substrates or cofactors to regulate chromatin modification and gene expression. On the other hand, epigenetic dysregulation mediates a unique metabolic microenvironment within tumors, which together are involved in tumor progression and treatment resistance [154].

### 4.1. DNA Methylation and Tumor Metabolism in HCC

The significant association between hepatic DNA methylation and IR in patients with MASLD underscores the potential mechanism by which MASLD may develop and worsen over time [155]. It is noteworthy that genes related to specific metabolic pathways exhibited hypermethylation and decreased expression levels, while numerous genes involved in tissue repair displayed hypomethylation and increased expression levels, indicating a potential dysregulation in lipid metabolism, oxidative stress response, fibrogenesis, and possibly even carcinogenesis [156]. GNMT regulates glucose homeostasis, and GNMT deficiency may have downstream effects on various metabolic pathways. Analyzing metabolism revealed heightened lipid buildup, polyamine creation and degradation, and transsulfuration in GNMT-deficient mice. This suggests that the absence of GNMT results in metabolic reprogramming, shifting carbons away from gluconeogenesis toward pathways utilizing S-adenosylmethionine (SAM) [157].

Deficiency of carbamoylphosphate synthase 1 (CPS1), an enzyme in the urea cycle, leads to an increase in ammonia levels and triggers the activation of FAO. This process generates ATP, which supports the rapid proliferation of HCC cells [158]. Notably, CPS1 was discovered to undergo hypermethylation in HCC, which is associated with a decrease in CPS1 mRNA expression. Hypermethylated transcription-repressed genes related to ureagenesis and amino acid metabolism have been observed in patients with MASLD. The findings from both experimental studies and human research support the idea that methylation of specific gene promoters can impact the functioning of the urea cycle [159].

### 4.2. Histone Modification and Tumor Metabolism in HCC

Nuclear factor erythroid 2-related factor 2 (Nrf2) is a crucial transcription factor controlling the response to oxidative stress, and it is involved in metabolic reprogramming in various types of cancer. Specifically, in the absence of Nrf2, there is a decrease in acetyl-CoA production, which in turn leads to a reduction in histone acetylation within tumors [160]. Zhao and colleagues identified that ubiquitin protein ligase E3 component N-recognin 7 (UBR7) plays a protective function in the development of HCC by hindering metabolic reprogramming toward aerobic glycolysis [161]. UBR7 is involved in monoubiquitinating histone H2B and acting as a histone chaperone for post-nucleosomal histone H3 [162,163]. Malic enzyme 1 (ME1) is an NADP (+)-dependent enzyme that links glycolysis and the TCA cycle. Using immunoprecipitation and mass spectrum data, Fu and colleagues demonstrated that deacetylation of phosphoglycerate mutase 5 boosts the activity of the ME1, thereby stimulating lipid synthesis and the proliferation of liver cancer cells [164]. Metabolic rewiring of glycoxenogenic enzyme phosphoenolpyruvate carboxykinase 1 (PCK1) disrupts hexosamine synthesis pathway-mediated O-GlcNAcylation and induces cataplerosis in the TCA cycle [165,166]. Gou et al. recently reported that PCK1 stimulates the synthesis of SAM via the serine synthesis pathway [167]. The PCK1-dependent generation of SAM boosts H3K9me3 modification on the promoter of S100A11, leading to the downregulation of the PI3K/AKT signaling pathway and ultimately suppressing the progression of HCC.

The unique structure of the histone variant macroH2A1 may assist cancer cells in cell cycle regulation, as well as DNA repair and transcription [168]. macroH2A1 successfully altered the metabolism of carbohydrates and lipids in HCC cells to promote the transformation into cancer stem cells by increasing lipid accumulation via activation of the LXR pathway [169].

Histone lactylation is a newly discovered histone modification that involves the addition of a lactyl group to the lysine residue of histones [170]. This modification has been found to have a positive association with glycolytic rate and show unique changes over time compared with histone acetylation. Interestingly, histone lactylation may induce gene transcription via E1A-binding protein (p300) and P53, thereby stimulating macrophage transition to the late-phase M2 phenotype.

### 4.3. ncRNA and Tumor Metabolism in HCC

The NcRNA-mediated reprogramming of FA metabolism is a key process from MASLD to HCC [171]. Numerous ncRNAs can directly control the activity of enzymes that synthesize lipids in liver cancer cells, leading to DNL. Of these, FASN and SCD are the most extensively researched enzymes. LncRNA ARSR has been identified as a promoter of FA accumulation, cell proliferation, and invasion in hepatocytes as a result of stimulating FASN and SCD activity by upregulating Yes-related protein 1 [172]. A newly discovered oncogenic lncRNA, RP11-386G11.10, functions as an endogenous RNA for miR-345-3p, regulating the expression of downstream lipogenic enzymes like FASN, which results in lipid accumulation within HCC cells [173]. Liu and colleagues additionally discovered that lncSNHG6 activates the interaction between mTOR and fas-associated factor family member 2 at ER-lysosome contacts and enhances the recruitment of mTORC1 to lysosomes in a cholesterol-dependent manner, further promoting the transition from MASLD to HCC [125]. Additionally, some lncRNA (Tacc1, lnc027912, Mef2c) are also involved in the reprogramming of lipid metabolism, such as mTOR/AMPK/SREBP 1c regulation in MASLD [174,175]. In addition, ncRNA can regulate the expression of FA transporter-related proteins. As an illustration, the inhibition of fatty acid-binding protein 1 (FABP1) expression by miR-603 contributes to the promotion of FA metabolism and synthesis-related proteins. This, in turn, elevates cellular oxidative stress levels, ultimately facilitating the metastasis of HCC [176].

Several miRNAs were reported to regulate glycolysis. This is, in fact, true for miR-125b, which targets HK2, or miR-34a, which is lactate dehydrogenase (LDHA), an essential enzyme in glycolysis, thereby restraining glycolysis in HCC cells [177,178]. The downregulation of miR-122 in MASH targets PKM2, resulting in enhanced glycolysis [179]. Moreover, factors beyond genetic mutations also play a role in the activation of the RAS/RAF/MEK/ERK pathway in HCC. Epigenetic mechanisms may contribute to tumor progression. Several miRNAs have been identified as potential activators of the RAS/RAF/MEK/ERK pathway in HCC [180]. The tumor-promoting factor SMYD3, targeted by miR-346, is significantly upregulated to trigger MAPK signaling, which facilitates the development of RAS-driven HCC and leads to the methylation of MAP3K2 [181].

In addition, circRNA MAT2B functions as a decoy for miR-338-3p, promoting the expression of PKM2 and increasing aerobic glycolysis, which contributes to the progression of HCC in hypoxic conditions [182]. A novel identified circRNA, circRHBDD1, has been found to enhance aerobic glycolysis and limit the efficacy of anti-programmed death-1 (PD-1) therapy in HCC [183]. LncRNAs are also involved in regulating the expression of transporter proteins and metabolic enzymes in glucose metabolism and regulate aerobic glycolysis through signaling pathways such as LKB1/AMPK, HIF, etc. The activation of RAS signaling pathways and associated lncRNAs are early molecular events in the reprogramming events [184]. lncRNAs can encode some microproteins. The microprotein KRASIM interacts with and co-localizes with the KRAS protein in the cytoplasm of HCC cells. The overexpression of KRASIM reduces the level of KRAS protein, leading to the inhibition of the ERK signaling pathway in HCC cells [185]. Chen et al. found that overexpressed lncRNASNHG6 in HCC cells binds to block proliferation 1 (BOP1) proteins to promote glycolysis and cancer cell proliferation while inhibiting apoptosis [186]. Xu and colleagues discovered that exosomes derived from tumor-associated macrophages (TAMs) have a significant impact on the regulation of glucose metabolism and cell proliferation within HCC cells [187]. Specifically, they found that TAMs release M2 macrophage polarization-associated lncRNA via exosomes to tumor cells to enhance the stability of aldehyde dehydrogenase 1A3, thereby promoting aerobic glycolysis and proliferation in HCC cells.

### 4.4. m6A Modification and Tumor Metabolism in HCC

In the m6A methylome of mice with MASLD, genes associated with lipid metabolism displayed marked hypermethylation. The differential m6A methylation could potentially affect lipid metabolism-related genes through RNA splicing factors, ultimately impacting the regulation of lipid metabolism [188]. Through protein expression analysis, immunoprecipitation-qPC, and RNA sequencing, Yang et al. demonstrated that the upregulated METTL14 interacts with the mRNA of ACLY and SCD1, causing a shift in their expression profiles [36]. This alteration ultimately exacerbates FA synthesis and lipid accumulation, contributing to the advancement of MASLD and HCC. Additionally, Pan and colleagues showed that the m6A writer METTL3 triggers the m6A-SCAP-cholesterol pathway, resulting in the suppression of anti-tumor CD8+ T cells, subsequently facilitating the development of MASLD-HCC [72]. Recently, METTL5 has been demonstrated to enhance cell proliferation and aggrandize FA metabolism by impacting both the FAO and DNL pathways [189]. These findings highlight a crucial regulatory process in which m6A modification influences lipidomics in MASLD and HCC.

FTO, an important m6A demethylase, is increased in MASLD liver patients, leading to decreased FAO and increased lipid accumulation [190]. Furthermore, FTO activates the SREBP/cell death-inducing DFF45-like effector C signaling pathway in an m6A-dependent manner and increases lipid accumulation in HCC [191].

In addition, the m6A reader YTHDC2 can balance the hepatic lipid metabolism by modifying the mRNA of lipid metabolism-related genes [77]. IGF2BP2, another distinct reader of m6A, could potentially intervene in TAG accumulation by influencing the degradation of CPT1A and PPARa mRNA [192]. Further research has validated that dysregulation of IGF2BP2 is associated with the progression of MASLD to HCC [193].

5-methylcytosine (m5C) is an important mRNA modification as well. Nucleolar protein 2 (NOP2), also known as NSUN1, is an M5C methyltransferase [194]. Zhang et al. identified that NOP2 relies on m5C modification to maintain c-Myc stability, leading to the Warburg effect in HCC cells [195]. C-Myc, an oncoprotein, is responsible for activating nearly all genes related to glycolysis and is crucial for regulating glycolysis in normal oxygen levels [196].

## 5. Therapeutic Potential of Targeted Epigenetic Modifiers and Metabolic Reprogramming in HCC

### 5.1. Pre-Clinical Studies

#### 5.1.1. Epigenetic Targets

Epigenetic drugs are a group of compounds that target disturbed epigenetic changes in different disease states. The role of epigenetic drugs in liver disease management is most clearly defined in the field of HCC. These drugs are mainly inhibitors of epigenetic-related enzymes, including DNMT, histone methyltransferase (HMT), histone demethylase (HDM), HAT, HDAC, etc. [7] (Figure 4).

Experimental therapies focusing on DNA methylation-specific mechanisms in MASLD and HCC have been emerging [197]. Targeting aberrant DNA methylation using DNMT inhibitors (DNMTis) is currently being explored. Liu et al. introduced a novel DNA methylation inhibitor, Guadecitabine (SGI-110), that has demonstrated promising anti-proliferative effects on HCC cell lines [198]. Moreover, SGI-110 effectively reversed the silencing of endogenous retroviruses in liver cancer cells, which in turn boosted the immune system’s response to liver cancer and could be utilized to enhance the sensitivity of immune checkpoint inhibitors in organisms [199].

The efficacy of HDAC inhibitors (HDACis) in experimental HCC has been shown through various studies [200]. Illustratively, panobinostat, a non-selective pan-HDACi, has exhibited the competence to trigger apoptosis, reprogram cancer cell metabolism, and mitigate tumor angiogenesis [201]. In HCC, bromodomain and PHD finger containing-1 (BRPF1) induce the activation of oncogenes by stimulating gene promoter H3K14 acetylation, and BRPF1-targeted inhibitor GSK5959 demonstrates a promising capacity to ameliorate tumor progression in murine models of HCC [202]. The integration of oncolytic and epigenetic therapies is a promising strategy for the management of multiple cancers [203]. Telomelysin, a telomerase-specific oncolytic adenovirus, and AR42, an HDACi, have shown anti-cancer properties in pre-clinical studies involving human HCC [204].

The development of specialized drugs for histone demethylation in HCC is equally appealing and encouraging [205]. Epigenetic inhibitors targeting JmjC lysine HDM, including JIB-04, GSK-J4, and SD-70, have been reported to attenuate HCC aggressiveness and viability [206] and, more notably, protect against the progression of experimental MASH-related HCC [207].

In addition, ncRNAs are important epigenetic mediators in the progression of multiple HCCs, which are key targets for effective cancer therapy [208]. Bergamini et al. identified miR-494 as a metabolic driver of HCC cells to a glycolytic phenotype and established the potential of antimiR-494 oligonucleotides used in conjunction with sorafenib and metabolic agents [209]. Significant advancements have been made in the optimization of delivery strategies and chemical modification techniques for ncRNA targets such as exosomes, lipid particles, and polymers in recent years [210]. These approaches targeting ncRNAs are more straightforward than traditional methods that focus on protein-binding inhibitors. For instance, administering adenoviral vectors carrying the tumor suppressor lncRNA PRAL significantly decelerates the progression of HCC in mice [211]. Zuo et al. developed a new PLGA PEG nanoplatform encapsulating si-LINC00958 for treating HCC [73]. This platform displayed accurate tumor-targeting, cellular drug uptake, and controllable release. Parallel to this, the assessment of circulating miRNA profiles is expected to provide a non-invasive tool to evaluate and supervise the severity of HCC [212].

In recent years, pre-clinical studies targeting m6A-specific modifications have been emerging. Zhang et al. found that lipid nanoparticles targeting the m6A reader YTHDF1 could inhibit stemness and augment sensitivity to targeted therapies in HCC cells, thereby enhancing the potency of sorafenib and lenvatinib in vivo [213]. Furthermore, Pan et al. found that nanoparticle siMETTL3 or METTL3 specific inhibitor STM2457 can elicit a profound anti-tumor immune response, and targeting m6A writer METTL3 may induce MASLD-HCC tumor regression, especially when combined with anti-PD-1 therapy [72].

Currently, the application of non-oncology agents for novel cancer treatments is gathering steam [214]. Wang et al. demonstrated that the combination of the non-oncology agent meticrane, which is typically prescribed for essential hypertension, and the epigenetic drug (5AC/DNMT1) inhibited the viability of liver cancer cells more effectively than either drug alone [215].

#### 5.1.2. Targeting Metabolic Reprogramming in HCC

Various research studies have concentrated on inhibiting enzymes in the FA biosynthesis pathway to hinder the growth of HCC cells [107]. Specifically, enzymes such as acetyl-CoA carboxylase (ACC), SCD, and FASN have been targeted through pharmacological means to impede lipid synthesis. Researchers have shown in a pre-clinical study that FASN inhibitors may potentially enhance the effectiveness of HCC therapies [216]. Building on this finding, a more recent study by Shueng and colleagues showed that Orlistat’s ability to inhibit FASN could address resistance to targeted drug therapy by disrupting HCC’s metabolic reprogramming [217]. Additionally, targeting upstream transcription factors could be a promising approach for developing therapeutic interventions for MASH-HCC. Obeticholic acid was found to function as an agonist for the farnesoid X receptor, thereby impacting FA metabolism to restrict the onset and development of MASH-HCC [218].

Saturation of NADH shuttles fuels aerobic glycolysis in cancer cell proliferation, according to a new study by Wang et al. [219]. Consequently, metabolic therapy for HCC entails dual targeting of Warburg effects and oxidative phosphorylation. The PKM2 nuclear translocation appears to be critical for activating aerobic glycolysis in HCC, while PKM1 steers metabolism toward oxidative phosphorylation [220]. By targeting PKM splicing and favoring the expression of the PKM1 isoform, antisense oligonucleotide treatment effectively inhibits HCC dependence on aerobic glycolysis, leading to a reduction in HCC cell proliferation [221]. Furthermore, protein arginine methyltransferase 3 (PRMT3) is a key driver of glycolysis in HCC cells. The PRMT3-specific inhibitor SGC707 prevents PRMT3-mediated LDHA methylation and indirectly suppresses glycolysis, as well as HCC progression [222]. Targeting glucose transporter1 (Glut1) has been investigated for HCC treatment. For example, BAY-876 is a Glut1 antagonist, and a single injection of microcrystalline BAY-876 into HCC tumor tissues led to the inhibition of glucose uptake, proliferation, and EMT of HCC [223]. Li et al. identified that Ilicicolin H can target phosphoglycerate kinase 1, a highly expressed enzyme in HCC cell lines, which inhibits the lactate production and glucose uptake of HCC cells [224]. Several natural compounds have demonstrated significant anti-HCC properties by targeting glycolysis genes or proteins. Oleuropein, for example, can hinder HCC cell glycolysis through glucose-6-phosphate isomerase inhibition, resulting in potent anti-tumor effects in animal models without any adverse side effects [225]. Erianin effectively suppresses pyruvate carboxylase enzyme activity, impairs glycolysis, and induces oxidative stress, resulting in a shortage of needed energy for the growth of HCC cells [226]. Deoxyelephantopin was found to have an impact on key metabolic processes in HepG2 cells, specifically suppressing glycolysis and reducing glucose uptake and lactic acid production, and all of the influences through the PI3K/Akt/mTOR/HIF-1α signaling pathway [227]. Recently, Wu et al. found that HuaChanSu has anti-HCC properties by inhibiting G6PD in PPP [228]. The compound is expected to inhibit PPP flux by suppressing NADPH production and reducing nucleotide levels, providing a potential therapeutic approach for the treatment of HCC.

Metformin is a well-known medication commonly used to manage individuals with type 2 diabetes mellitus [229]. Recent studies have found that metformin not only mitigates the Warburg effect by inhibiting phosphofructokinase-1 [230] but also promotes FAO [231], demonstrating promising anti-tumor effects in pre-clinical studies of HCC. In addition to metformin, other classes of antidiabetic medications have also been considered for their potential benefits in treating HCC. The ability of sodium-glucose cotransporter 2 inhibitor Canagliflozin to inhibit glucose uptake in HCC cells expands its potential applications beyond diabetes management, offering promise for patients with HCC [232]. CP-91149 is a non-insulin-dependent diabetic drug that targets glycogen phosphorylase [233]. Barot et al. found that CP-91149 inhibits glycogenolysis, interfering with glycolysis and the PPP and causing mitochondrial dysfunction in HepG2 cells [234]. Recently, Syamprasad et al. indicated that the aldose reductase inhibitor epalrestat and its analog NAR1-29 could be combined with antidiabetic therapy to combat diabetes-induced MASLD and even HCC [65].

Concerning amino acid metabolism, the development of novel therapeutics against HCC is expected to target the metabolic vulnerability of glutamine-dependent HCC. Although the glutaminase inhibitor CB-839 had limited effect on HCC, it induced the apoptosis of HCC and inhibited HCC xenografts in mice when combined with V-9302, a novel inhibitor of glutamine transporter alanine-serine-cysteine transporter 2 [235].

### 5.2. Clinical Trials

#### 5.2.1. Epigenetic Drugs

Clinical validation of epigenetic drugs for HCC is currently underway based on the experimental evidence described above. A few clinical trials have demonstrated that epigenetic drugs can improve the sensitivity of tumors to chemotherapy, prompting further investigation of epigenetic inhibitors in combination with conventional chemotherapeutic agents [236].

SGI-110 is a second-generation DNMTi that concluded a phase II clinical trial in 2020 (NCT01752933) in combination with sorafenib and oxaliplatin for HCC. Similarly, another DNMTi, called decitabine, showed a positive clinical response and a safe toxicity profile for advanced HCC patients in phase II clinical trials [237]. Encouragingly, decitabine and an HMT inhibitor targeting G9a (BIX-01294) demonstrated a dual inhibitory effect and showed strong potential for HCC treatment in the clinic [238].

Belistat, an HDACi, successfully prevented further tumor progression in Phase I/II clinical trials for the treatment of patients with unresectable HCC [239]. In addition, the preliminary efficacy of the pan-HDACi resimonstat in conjunction with sorafenib has been demonstrated in the treatment of advanced HCC [240]. Further research and larger-scale clinical trials are warranted to validate these initial findings and determine the optimal dosing and treatment regimens for maximizing the therapeutic benefits of HDAC inhibitors in HCC patients.

#### 5.2.2. Targeting Metabolic Reprogramming in HCC

Currently, treatments targeting FA anabolism are leading the way, particularly those directed against FASN, which are in clinical trials. TVB-2640 is the first FASN inhibitor to undergo clinical studies and has demonstrated positive clinical responses in patients with ovarian and breast cancer [241]. The utilization of a TVB-2640 in a clinical trial for MASH demonstrated effectiveness in the reduction of liver fat and improvement of biochemical biomarkers [242]. A phase II clinical trial of PF-5221304 for the treatment of patients with fibrosing MASH (NCT04321031) concluded with a favorable efficacy and safety profile. In light of the excellent performance of ACC inhibitors in regulating liver metabolism, it has been hypothesized that PF-05221304 may exert a therapeutic effect on MASH-induced HCC or increase the sensitivity of tumor cells to immunotherapy.

In the battle against cancer challenges, researchers have focused on targeting glucose metabolism by primarily using agents that block glycolysis in HCC treatment. It is important to highlight that the phase I clinical trial for the treatment of HCC using oroxylin A has been authorized by the National Medical Products Administration (NMPA; ChiCTR2100051434). Studies revealed that oroxylin A targets transketolase directly to inhibit non-oxidative PPP and trigger p53 signaling, resulting in anti-cancer activity [243]. In 2022, Olutasidenib (FT-2102), a selective inhibitor of mutant isocitrate dehydrogenase 1, was approved by the U.S. Food and Drug Administration to treat relapsed/refractory acute myeloid leukemia. A completed phase Ib/II study (NCT03684811) has shown that Olutasidenib has some therapeutic activity and safety in the treatment of patients with advanced solid tumors, including HCC (NCT03684811). These approaches aim to leverage the synergistic effects of these agents in targeting glucose metabolism in cancer cells, thereby potentially enhancing the therapeutic outcomes for patients. All of these studies prompt deeper consideration of how a combination of targeted therapies for metabolism and dietary interventions might maximize anti-cancer effects, optimize HCC treatment, and improve patient prognosis. Agents targeting deregulated metabolism and epigenetics for the treatment of MASLD-HCC in pre-clinical or in-clinical trials have been summarized in Table 4.

## 6. Conclusions and Future Perspectives

In tumor cells, epigenetic modifications and metabolic reprogramming are highly intertwined. The close interaction between the two is similarly manifested in the progression of MASLD to HCC. Ahmed et al. [110] demonstrated significant differences in pathway regulation during MASH progression to HCC, including the downregulation of FA metabolism, biogenic amine synthesis, mTOR signaling, PPAR-α-related lipid metabolism, and amino acid metabolism. Conversely, upregulated signaling pathways in MASH-HCC patients involved DNA repair, BA metabolism, cholesterol metabolism, and methylation pathways. A deep understanding of the relationship between epigenetic modifications and metabolomics, as well as the purposeful treatment of relevant targets, is of great relevance for the development of targeted drugs.

Quantitative systems pharmacology (QSP) model-based drug transformation research is an emerging international cutting-edge drug development paradigm [244]. Several mature QSP models have been developed and put into the study of glucose metabolism and the Warburg effect, changing the shortcomings of previous clinical pharmacological models lacking mechanisms [245]. Various emerging technologies based on genomics and metabolomics have also come to the fore in the study of tumor heterogeneity, cancer clone evolution, and hepatocyte network, which are undoubtedly powerful tools for understanding MASLD-HCC metabolic reprogramming deeply [246,247]. More importantly, it is essential to consider the impact of metabolites on different cell types within the liver cancer microenvironment.

The Barcelona Clinic Liver Cancer (BCLC) system is currently the most widely used staging system for liver cancer worldwide and has been endorsed by many guidelines, including the European Organization for Research and Treatment of Cancer. Its unique advantage lies in its comprehensive consideration of the general condition of the patient, tumor status, liver function, and the preferred treatment method according to different stages [248] (Figure 5). Note that the absence of etiology-specific clinical practice guidelines for HCC currently reflects the complex nature of HCC and the challenges in formulating tailored approaches for different patient populations [249]. Since MASLD is associated with liver cancer risk, proper caloric restriction, weight loss, and diabetes management are important strategies for liver cancer prevention; proper nutrition is equally important in the management of HCC. By ensuring that patients are receiving enough calories and nutrients, healthcare professionals can help improve treatment outcomes and overall quality of life for individuals battling HCC [250]. However, the majority of MASLD-HCC is diagnosed at an advanced stage, making systemic therapy the only option [251,252]. Although immunotherapy has been approved for the treatment of MASLD-related HCC, emerging evidence suggests that this specific subset of patients may not respond well to this treatment option [253]. Emerging fields in the treatment of MASH-HCC include vaccination with peptides or DNA, adoptive transfer of immune cells, and monoclonal antibodies against PD-1, etc. [119]. These strategies are undergoing further investigation to determine their effectiveness in combination with existing standard treatments for metabolic-associated HCC. Interestingly, some drugs and natural products used to treat MASLD may also exhibit anti-HCC effects. For instance, glucagon-like peptide 1 receptor agonist (GLP-1 RA) is a novel medication for diabetes that has demonstrated efficacy in enhancing liver function and diminishing hepatic fat levels in individuals suffering from MASLD [254]. The effectiveness of GLP-1 RA has also been positively demonstrated in HCC mice [255]. However, further research is needed to determine if the metabolic effects of GLP-1 RA are solely responsible for these benefits or if its anti-inflammatory properties also play a role. Compounds such as gastrodin, curcumin, genistein, and silymarin have demonstrated significant efficacy against MASLD and also exhibited anti-HCC activity in vivo and in vitro [256]. Ultimately, bridging the gap between pre-clinical research and clinical application will be crucial in harnessing the full therapeutic potential of these natural compounds for the treatment of HCC.

Given that HCC may manifest in MASLD patients before cirrhosis onset, balancing necessary early diagnosis with unnecessary invasive testing is one of the major challenges in the current management of HCC [257]. State-of-the-art epigenetic techniques have the potential to pinpoint the exact epigenetic modifications associated with MASLD and HCC, improving the accuracy and timeliness of disease diagnosis. Furthermore, researchers can develop innovative methods for diagnosing cancer without invasive procedures by exploiting HCC cells’ dependence on metabolic reprogramming [258]. Previous research has primarily focused on the role of miRNAs in regulating HCC metabolism while overlooking the involvement of lncRNAs and circRNAs in the intricate regulation of multiple genes, among them miRNAs, in HCC. It is imperative to conduct studies that explore the comprehensive biological significance of ncRNAs and their capacity for reprogramming.

HCC treatment research has evolved from a focus on efficacy to an exploration of underlying mechanisms. However, current research still lacks breadth and depth due to limited scale, unclear targets, and insufficient clinical trials. To overcome these deficiencies, it is essential to seamlessly incorporate established research with cutting-edge approaches, such as leveraging artificial intelligence for high-throughput screening, integrating proteomics with network pharmacology, and conducting clinical events-based research. Therefore, a comprehensive understanding of the characteristic alterations of metabolic and epigenetic modifications in the body during tumorigenesis and development and a systematic summary of past findings, current trends, and future research directions in metabolism and epigenetics will help to develop new combined therapeutic strategies and ideas for targeting the characteristic metabolic-epigenetic alterations in tumors.

## Figures and Tables

**Figure 1 metabolites-14-00325-f001:**
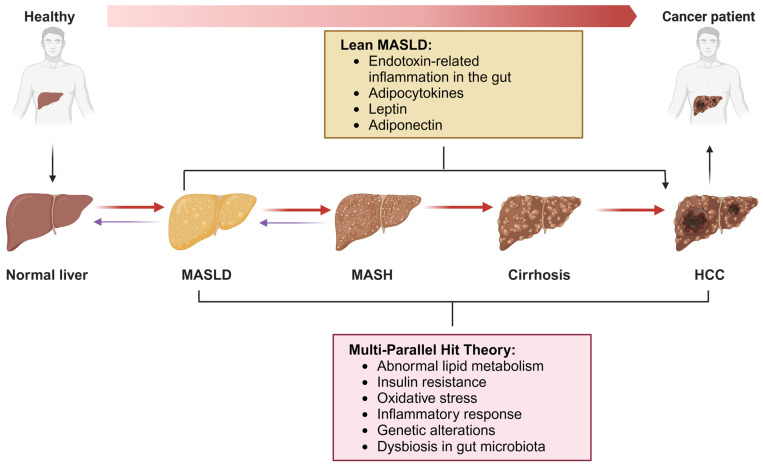
Pathogenesis spectrum of MASLD–HCC. Abbreviations: MASLD, metabolic dysfunction-associated steatotic liver disease; MASH, metabolic dysfunction-associated steatohepatitis; HCC, hepatocellular carcinoma.

**Figure 2 metabolites-14-00325-f002:**
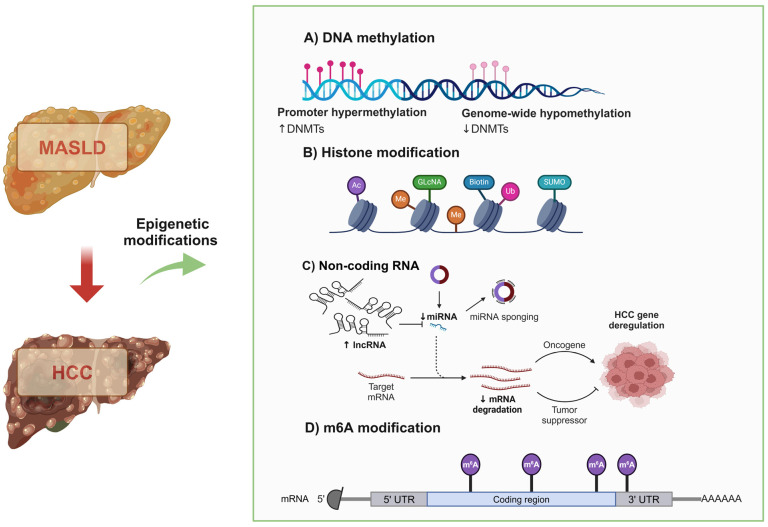
Several types of epigenetic modifications in MASLD-induced HCC. Abbreviations: DNMTs, DNA methyltransferases.

**Figure 3 metabolites-14-00325-f003:**
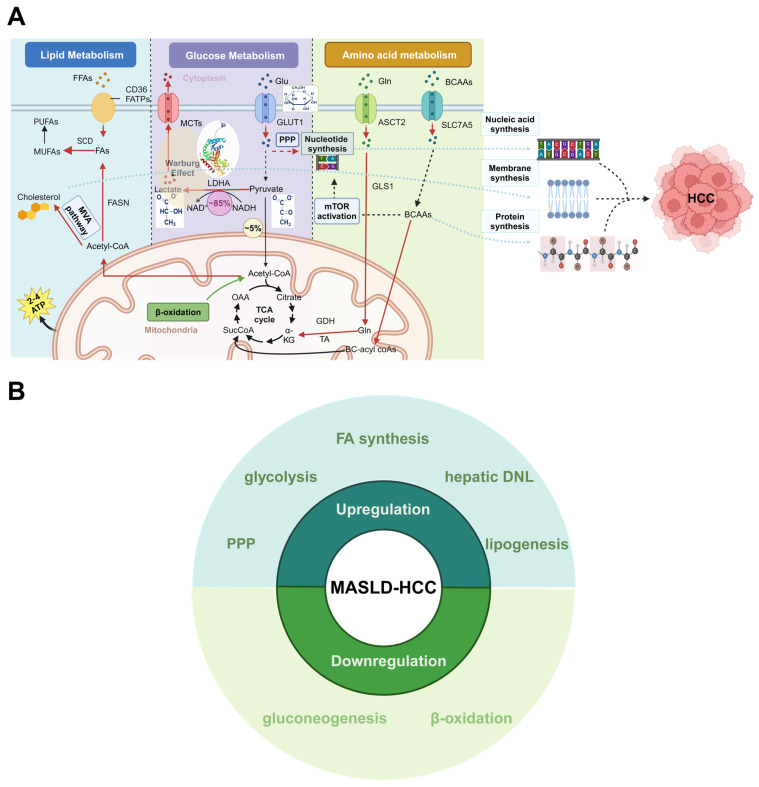
Three types of metabolic reprogramming (**A**) and dysregulation of metabolic pathways in MASLD-HCC (**B**). As depicted in the literature review, oncogenic pathways are triggered by alterations in various metabolic pathways at the onset of neoplasia. These modifications at the cellular and metabolic levels provide cancer cells with advantages to support their rapid growth and proliferation in response to the hostile tumor environment. Altered regulation of FA and cholesterol metabolism in HCC related to MASLD. Fatty acid β-oxidation is suppressed for adaptation to a lipid-rich environment. Acetyl-CoA is converted into FAs through lipogenesis. HCC cells demonstrate heightened glucose absorption and rely on aerobic glycolysis as their primary energy source. Rather than being oxidized in the mitochondria, pyruvate is predominantly converted into lactate. The increased glucose uptake also meets the substrate demands of the PPP. This is important for nucleotide biosynthesis and nucleic acid replication. Dysregulation of genes and metabolic intermediates involved in amino acid and glutamine metabolism occurs in HCC. This dysregulation can be triggered by aberrantly activated oncogenes and the loss of tumor suppressors, such as ncRNAs. Abbreviation: FFA, free fatty acids; FATPs, fatty acid transport proteins; MUFAs, monounsaturated fatty acids; PUFAs, polyunsaturated fatty acids; SCD, stearoyl-CoA desaturase; FAs, fatty acid; FASN, fatty acid synthase; Glu, glucose; MCTs, monocarboxylate transporters; GLUT1, glucose transporter 1; PPP, pentose phosphate pathway; LDHA, lactate dehydrogenase; Gln, glutamine; ASCT2, alanine-serine-cysteine transporter 2; BCAAs, branched-chain amino acids; SLC7A5, solute carrier family 7 member 5; GLS1, glutaminase 1; OAA, oxaloacetate; TCA cycle, tricarboxylic acid cycle; α-KG, α-ketoglutarate; GDH, glutamate dehydrogenase; TA, transglutaminase; DNL, De novo lipogenesis; PPP, pentose phosphate pathway.

**Figure 4 metabolites-14-00325-f004:**
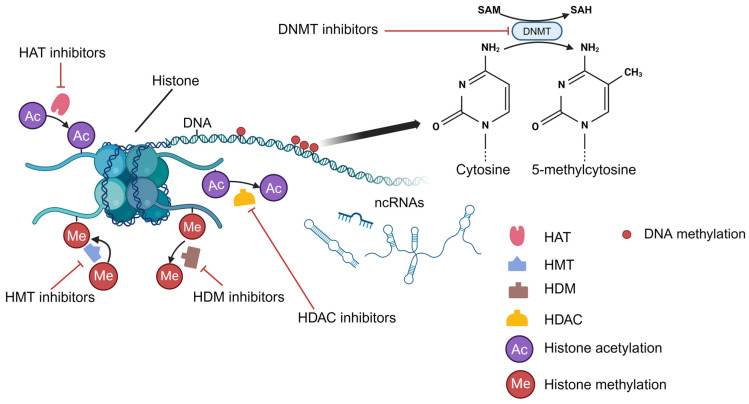
By targeting specific epigenetic modifications, researchers can develop a variety of small molecule inhibitors. Abbreviation: HAT, histone acetyltransferase; Ac, acetylation; Me, methylation; HMT, histone methyltransferase; HDM, histone demethylase; HDAC, histone deacetylase; SAM, s-adenosylmethionine; SAH, s-adenosine homocysteine; DNMT, DNA methyltransferase.

**Figure 5 metabolites-14-00325-f005:**
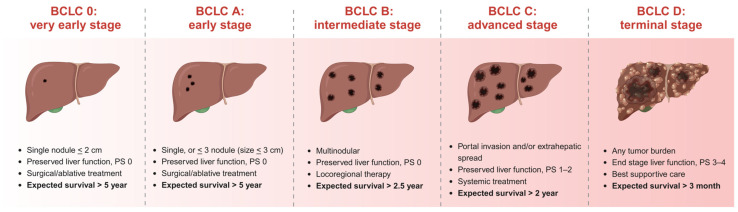
Barcelona Clinic Liver Cancer (BCLC) staging and classification.

**Table 1 metabolites-14-00325-t001:** Pathogenic mechanisms of MASLD-HCC progression.

Involved Pathogenic Mechanisms	Outcomes	References
Lipotoxicity	The molecular mechanism by which CD36 accelerated the progression of HCC was to promote the expression of AKR1C2 and thus enhance FA intake	[17]
Oxidative DNA damage	Oxidative stress and 8-OHdG formation in the DNA in mice liver cells are the important characteristics of MASH-associated hepatocarcinogenesis	[19]
Mitophagy	Hepatic mitochondrial depolarization occurs early in mice fed a Western diet, followed by increased mitophagic burden, suppressed mitochondrial biogenesis and dynamics, and mitochondrial depletion, which ultimately contributes to the progression of MASH toward HCC	[22]
Genetic factor	A rare genetic variant in the gene MTTP has been identified as responsible for the development of progressive MASLD in a four-generation family with no typical disease risk factors	[25]
Genetic factor	Mutational profiling of MASH-HCC tumors revealed TERT promoter (56%), CTNNB1 (28%), TP53 (18%), and ACVR2A (10%) as the most frequently mutated genes. ACVR2A mutation rates were higher in MASH-HCC than in other HCC aetiologies (10% vs. 3%, *p* < 0.05)	[26]
Genetic factor	Somatic mutations in MASH mice did not reveal increased tumorigenesis	[27]
Dysbiosis in gut microbiota	Dietary cholesterol drives NAFLD-HCC formation by inducing the alteration of gut microbiota and metabolites in mice	[28]
Other mechanisms	The activation of the RAS/RAF/MEK/ERK pathway may contribute to HCC development; KRAS activation downstream of c-Met is sufficient to induce clinically relevant HCC in cooperation with mutant β-catenin.	[30,33]
Other mechanisms	The dynamic shift in HSC subpopulations and their mediators during MASLD is associated with a switch from HCC protection to HCC promotion	[35]

Abbreviation: AKR1C2, aldo-keto reductases family 1 member C2; FAs, fatty acids; 8-OHdG, 8-hydroxy-2′-deoxyguanosine; MTTP, microsomal triglyceride transfer protein; TERT, telomerase reverse tranase; CTNNB1, beta-catenin; ACVR2A, activin receptor type 2A; RAS, rat sarcoma virus; RAF, rapidly accelerated fibrosarcoma; MEK, mitogen-activated protein kinase; ERK, extracellular signal-regulated kinase; HSC, hepatic stellate cell.

**Table 4 metabolites-14-00325-t004:** List of some potential anti-cancer drugs targeting epigenetics and metabolism.

Agents	Developmental Stage	Functions in HCC	Target	Reference/Clinical Trial Number
Epigenetic drugs				
Guadecitabine (SGI-110)	Pre-clinical	Anti-proliferative effects on HCC cell lines; enhances the sensitivity of immune checkpoint inhibitors in organisms	DNMT	[198,199]
Panobinostat	Pre-clinical	Triggers apoptosis, reprograms cancer cell metabolism, and mitigates tumor angiogenesis	HDAC	[201]
GSK5959	Pre-clinical	Suppresses HCC cell growth	BRPF1	[202]
AR42 + Telomelysin	Pre-clinical	AR42 reduced Telomelysin-induced phospho-Akt activation and enhanced Telomelysin-induced apoptosis	HDAC	[204]
JIB-04, GSK-J4, SD-70	Pre-clinical	Attenuates HCC aggressiveness and viability	JmjC lysine HDM	[206,207]
AntimiR-494 oligonucleotides + Sorafenib	Pre-clinical	Inhibition of HCC cells to a glycolytic phenotype	miR-494	[209]
lncRNA-PRAL	Pre-clinical	Inhibits HCC growth and induces apoptosis	p53	[211]
PLGA-based nanoplatform encapsulating LINC00958 siRNA	Pre-clinical	Inhibits HCC lipogenesis and progression	LINC00958	[73]
STM2457	Pre-clinical	Elicits a profound anti-tumor immune response	METTL3	[72]
5AC + Meticrane; CUDC-101 + Meticrane; ACY1215 + Meticrane	Pre-clinical	Inhibited the viability of liver cancer cells	DNMT1; HDACs; HDAC6	[215]
Guadecitabine (SGI-110) + Sorafenib + Oxaliplatin	Phase II	-	DNMT	NCT01752933
Decitabine	Phase II	-	DNMT	[237]
BIX-01294 + Decitabine	-	-	HMT G9a + DNMT	[238]
Belistat	Phase I/II	-	HDAC	[239]
Resimonstat + Sorafenib	Phase I/II	-	HDAC	[240]
Agents targeted metabolism				
TVB3664	Pre-clinical	Ameliorates the fatty liver phenotype in the aged mice and AKT-induced hepatic steatosis	FASN	[216]
Orlistat	Pre-clinical	Disrupts HCC’s metabolic reprogramming	FASN	[217]
Obeticholic acid	Pre-clinical	Attenuates the development and progression of NASH-dependent HCC, possibly by interfering with SOCS3/Jak2/STAT3 pathway	FXR	[218]
SGC707	Pre-clinical	Prevents PRMT3-mediated LDHA methylation and indirectly suppresses glycolysis, as well as HCC progression	PRMT3	[222]
BAY-876	Pre-clinical	Inhibits glucose uptake, proliferation, and EMT of HCC	Glut1	[223]
Ilicicolin H	Pre-clinical	Inhibits the lactate production and glucose uptake of HCC cells	PGK1	[224]
Oleuropein	Pre-clinical	Inhibits HCC cell glycolysis	G6PI	[225]
Erianin	Pre-clinical	Impairs glycolysis and induces oxidative stress	PC	[226]
Deoxyelephantopin	Pre-clinical	Inhibits glycolysis and reduces glucose uptake and lactic acid production	PI3K/Akt/mTOR/HIF-1α pathway	[227]
HuaChanSu	Pre-clinical	Inhibits PPP flux	G6PD	[228]
Metformin	Pre-clinical	Mitigates the Warburg effect and promotes FAO	PFK1	[230,231]
Canagliflozin	Pre-clinical	Inhibits glucose uptake in HCC cells	SGLT2	[232]
CP-91149	Pre-clinical	Inhibits glycogenolysis, interfering with glycolysis and the PPP	PG	[234]
Epalrestat/NAR1-29	Pre-clinical	Combats diabetes-induced MASLD and even HCC	AR	[65]
CB-839 + V-9302	Pre-clinical	Targets the metabolic vulnerability of glutamine-dependent HCC	GLS and ASCT2	[235]
TVB-2640	Phase II	Reduces liver fat and improves biochemical biomarkers	FASN	[242]
PF-5221304	Phase II	-	ACC	NCT04321031
Oroxylin A	Phase I	Inhibits non-oxidative PPP and triggers p53 signaling	TK	ChiCTR2100051434
Olutasidenib (FT-2102)	Phase Ib/II	-	IDH1	NCT03684811

Abbreviations: DNMT, DNA methyltransferase; HDAC, histone deacetylase; BRPF1, bromodomain and PHD finger containing 1; HDM, histone demethylase; METTL3, methyltransferase 3; HMT, histone methyltransferase; FXR, farnesoid X receptor; PRMT3, protein arginine methyltransferase 3; LDHA, lactate dehydrogenase A; EMT, epithelial-to-mesenchymal transition; PGK1; phosphoglycerate kinase 1; SGLT2, sodium-glucose transport protein 2; PG, protein-glutaminase; AR, aldose reductase; GLS, glutaminase-1; ASCT2, alanine-serine-cysteine transporter 2; ACC, acetyl CoA carboxylase; TK, transketolase; IDH1, isocitrate dehydrogenase 1.

## Data Availability

No new data were created or analyzed in this study. Data sharing is not applicable to this article.

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
