# Peer review of "Crosstalk between Epigenetics and Metabolic Reprogramming in Metabolic Dysfunction-Associated Steatotic Liver Disease-Induced Hepatocellular Carcinoma: A New Sight"

_metabolites, 2024, doi:10.3390/metabo14060325_

Round 1

Reviewer 1 Report

Comments and Suggestions for Authors

Review of the article “Crosstalk between epigenetics and metabolic reprogramming in NAFLD-induced HCC: A new sight”

Currently, non-alcoholic fatty liver disease is one of the most common diseases in hepatology, leading to a deterioration in the quality of life, disability and death. This is primarily due to the high risk of progression to non-alcoholic fatty liver disease, liver failure and hepatocellular carcinoma. For many years, fatty liver disease was considered a relatively benign disease, often developing with type 2 diabetes mellitus, obesity, hyperlipidemia, and alcohol abuse. Non-alcoholic fatty liver disease is an independent nosological unit that combines clinical and morphological changes in the liver: steatosis, non-alcoholic steatohepatitis, fibrosis and cirrhosis. Non-alcoholic fatty liver disease is closely associated with obesity, especially abdominal obesity, and metabolic syndrome, which significantly increases cardiometabolic risk and affects the morbidity, prognosis and life expectancy of patients. Hepatic steatosis is a descriptive term characterizing the excessive accumulation of triglycerides in the cytoplasm of hepatocytes. Non-alcoholic steatohepatitis is a chronic diffuse liver disease, which is characterized by increased activity of liver enzymes in the blood and morphological changes in liver biopsies, similar to changes in alcoholic hepatitis; however, patients with nonalcoholic steatohepatitis do not drink alcohol in amounts that could cause liver damage.

The authors made a detailed review of this issue.They studied 235 literary sources for this. It is extremely important to note that careful analysis of data and their interpretation represent an essential link in the context of scientific knowledge.It is absolutely fair to note that this stage of the study allows us to identify the main patterns, trends and conclusions based on the studied facts.The process of in-depth analysis contributes to the development of logical conclusions, which significantly increases the authority and significance of the research.

I have a small note. NB.......It is unacceptable to use abbreviations in the title of an article and then explain what they are in the text. It's better to write it completely.

This will not interfere with my decision that the work can be accepted for publication. 

Author Response

Comments 1: It is unacceptable to use abbreviations in the title of an article and then explain what they are in the text. It's better to write it completely.

Response 1: Dear Professor, thank you for the title suggested, which improves the scientific quality of the article. The precedent version of the title has been replaced, becoming “Crosstalk between epigenetics and metabolic reprogramming in metabolic-associated fatty liver disease -induced hepatocellular carcinoma: A new sight”.

Reviewer 2 Report

Comments and Suggestions for Authors

A timely review article by Dr. Yang and the group elaborates on the role of epigenetics and metabolic reprogramming in HCC from a translational perspective. This review article is very well written and updated with the latest discoveries. However, a few things need to be addressed before it is ready for acceptance. They are as follows:

1. The Ras/Raf/MEK/ERK signaling pathway plays a significant role in the pathogenesis of HCC (PMID: 28454211), and RAS plays a significant role in metabolism and epigenetic changes (PMID: 33870211 and PMID: 26646588). 

Authors must discuss this based on HCC and how KRAS might be interconnecting metabolism and epigenetics, referring to the relevant works mentioned.

2. Add a chart/ table mentioning the ongoing clinical trials based on the metabolism and epigenetic biomarkers in HCC.

3. Similarly, add another table mentioning the drug's pre-clinical or in-clinical trials targeting metabolism and epigenetic biomarkers in HCC. 

4. Add a line mentioning the mutational landscape of HCC with oncogenic driver genes and tumor suppressors. This will give the readers an overall idea of HCC's genetic background. 

Author Response

Comments 1: The Ras/Raf/MEK/ERK signaling pathway plays a significant role in the pathogenesis of HCC. Authors must discuss this based on HCC and how KRAS might be interconnecting metabolism and epigenetics, referring to the relevant works mentioned.

Response 1: Dear Professor, we appreciate your valuable suggestions which have enhanced the scientific quality of the review. Following your suggestions and referencing related works[Refs. 30, 32, and 180], we have added a description of the Ras/Raf/MEK/ERK signaling pathway and KRAS to the text. [Page 3, paragraph 6, and lines 140-150; Page 12, paragraph 3, and lines 380-391; Page 13, paragraph 1, and lines 406-408; Page 14, paragraph 3, and lines 459-461; Page 19, paragraph 1, and lines 685-690; Page 19, paragraph 3, and lines 698-703]

Comments 2: Add a chart/ table mentioning the ongoing clinical trials based on the metabolism and epigenetic biomarkers in HCC.

Response 2: Dear Professor, thank you for your advice which makes this manuscript more scientific and better. We have extensively reviewed the relevant literature, but unfortunately, the available literature on this subject is quite limited. To fill this gap, we have added Tables 2, 3, and 4 to summarize the most recent research on metabolic and epigenetic biomarkers currently available in pre-clinical and clinical settings. In our future studies, we will devote more attention to this field to better understand the issue in more depth and detail. Thank you very much for your comments. [Pages 6-7; Pages 8-9; Pages 24-25.]

Comments 3: Similarly, add another table mentioning the drug's pre-clinical or in-clinical trials targeting metabolism and epigenetic biomarkers in HCC. 

Response 3: Dear Professor, we appreciate your valuable suggestions which have enhanced the scientific quality of the review. We have added Table 4 to provide a summary of drugs targeting metabolic and epigenetic biomarkers of HCC in pre-clinical and in-clinical trials. [Pages 24-25.]

Comments 4: Add a line mentioning the mutational landscape of HCC with oncogenic driver genes and tumor suppressors. This will give the readers an overall idea of HCC's genetic background. 

Response 4: Dear Professor, we appreciate your valuable suggestions which have enhanced the scientific quality of the review. We have added a description of the mutational landscape of HCC with oncogenic driver genes and tumor suppressors to the text. [Page 3, paragraph 4, and lines 126-135]

Reviewer 3 Report

Comments and Suggestions for Authors

In the review manuscript entitled “Crosstalk between epigenetics and metabolic reprogramming

 In NAFLD-induced HCC: A new sight”, the authors underscore the role the epigenetic mechanisms in driving the progression of MAFLD associated HCC and emphasize the difference between MAFLD and virus induced HCC and the role of metabolic reprogramming in the former. The review touches base on some very important aspects and molecular factors associated with the condition. However, there are a few shortcomings that should be corrected before accepting the manuscript for final publication.

The title needs to be changed from “Crosstalk between epigenetics and metabolic reprogramming In NAFLD-induced HCC: A new sight” to “Crosstalk between epigenetics and metabolic reprogramming in MAFLD-induced HCC: A new sight”. Also, NAFLD and NASH need to be corrected to MAFLD and MASH in accordance with the multinational liver societies’ 2023 recommendations.

Section wise comments:

Introduction:

Line 49: Change “which is mostly reversible and highly plasticity” to “which is mostly reversible and highly plastic.” The former statement is grammatically inaccurate.

Section 2.1.HCC tumorigenesis. This section provides a lot of information on how MASH progresses to HCC through metabolic reprogramming. Much of this information has been published previously and therefore, this section should be compressed into a more content-rich and length-reduced section. Please also consider including a table for referencing different mechanisms involved in the process for a easy flow of information and reader-friendliness.

Figure1. Both the figure and the figure legend are very generic, with little effort to allow the reader to focus on a core message. Please modify.

Section 2.2. Epigenetic dysregulation in the pathogenesis of NAFLD-induced HCC. The sub-sections included about DNA methylation, Histone modifications, NcRNAs, and m6A modification have been discussed vagely and on the surface. In order to do justice with the title of the manuscript, these sections should be discussed with more background, better connectivity and comprehensive information, substantiated by figures, tables and flow diagrams, for easier understanding.

Figure2. The figure is very generic and redundant, with these mechanisms being discussed innumerable times in published literature. Please modify.

Section 3. Metabolic shifts and reprogramming in NAFLD-induced HCC. The authors have discussed lipid, glucose and amino acid metabolism at length, but all of these have further not been individually and sequentially connected to epigenetics. It would be better if the authors focus on only one type of biomolecule class (lipid, glucose and amino acid metabolism) and provide more comprehensive details about the involvement of epigenetic mechanisms in this particular metabolic reprogramming.

Figure 3. The figure contains no meaning with regard to the graphical representation and the legend is not well explained. Please modify.

Line 855. Change metabonomics to metabolomics

Overall Comments. Overall, the manuscript would benefit from focusing on epigenetic mechanisms, their background and involvement in metabolic reprogramming and their identified and potential role in modifying metabolism to result in better patient outcomes in MAFLD induced HCC.

Comments on the Quality of English Language

Quality of English language is reasonable. A few typing errors or small misspellings detected which have been pointed out in the comments.

Author Response

Comments 1: The title needs to be changed from “Crosstalk between epigenetics and metabolic reprogramming In NAFLD-induced HCC: A new sight” to “Crosstalk between epigenetics and metabolic reprogramming in MAFLD-induced HCC: A new sight”. Also, NAFLD and NASH need to be corrected to MAFLD and MASH in accordance with the multinational liver societies’ 2023 recommendations.

Response 1: Dear Professor, we appreciate your valuable suggestions which have enhanced the scientific rigor of the review. The precedent version of the title has been replaced, becoming “Crosstalk between epigenetics and metabolic reprogramming in metabolic-associated fatty liver disease -induced hepatocellular carcinoma: A new sight”. Following your suggestions and referencing the multinational liver societies’ 2023 recommendations [Refs. 4], we have updated the terminology in the body of the text and the abstract from NAFLD and NASH to MAFLD and MASH. Here we did not list the changes but marked them in red in the revised paper.

Comments 2: Section 2.1.HCC tumorigenesis. This section provides a lot of information on how MASH progresses to HCC through metabolic reprogramming. Much of this information has been published previously and therefore, this section should be compressed into a more content-rich and length-reduced section. Please also consider including a table for referencing different mechanisms involved in the process for a easy flow of information and reader-friendliness.

Response 2: Dear Professor, thank you for your valuable suggestions, which have improved the scientific quality of the review. We have improved this section based on your valuable suggestions, deleted some clichés, and added information about mitochondrial autophagy, the mutational landscapes of oncogenic driver genes and tumor suppressor genes, as well as the important roles of the RAS/RAF/MEK/ERK signaling pathways in the pathogenesis of MAFLD-HCC. Furthermore, we have added Table 1 to summarize the various mechanisms. [Page 3, paragraphs 3-4, 6 and lines 114-135, 140-150; Page4]

Comments 3: Figure1. Both the figure and the figure legend are very generic, with little effort to allow the reader to focus on a core message. Please modify.

Response 3: Dear Professor, we appreciate your valuable suggestions which have enhanced the scientific quality of the review. We have modified Figure 1 to allow the reader to focus more on the pathological mechanisms by which MAFLD develops into HCC. [Page 5]

Comments 4: Section 2.2. Epigenetic dysregulation in the pathogenesis of NAFLD-induced HCC. The sub-sections included about DNA methylation, Histone modifications, NcRNAs, and m6A modification have been discussed vagely and on the surface. In order to do justice with the title of the manuscript, these sections should be discussed with more background, better connectivity and comprehensive information, substantiated by figures, tables and flow diagrams, for easier understanding.

Response 4: Dear Professor, thank you for your suggestion, which improves the scientific quality of the review. We have refined this section based on your valuable suggestions and added some references[Refs. 38-39, 49-50, 60-61, 63, 65-68, 73, 76-79] to provide a more comprehensive and coherent account of epigenetic dysregulation in the pathogenesis of MAFLD-induced HCC. In addition, we have added Table 2, Table 3, and Figure 2 for readers to understand. [Pages 6-7; Pages 8-9; Page 10]

Comments 5: Figure2. The figure is very generic and redundant, with these mechanisms being discussed innumerable times in published literature. Please modify.

Response 5: Dear Professor, we appreciate your valuable suggestions which have enhanced the scientific quality of the article. We have incorporated your valuable suggestions into the revised Figure 3, which is based on Figure 2 in the original manuscript. This new version provides the reader with an at-a-glance view of the three types of metabolic reprogramming in MAFLD-induced HCC. [Page 11]

Comments 6: Section 3. Metabolic shifts and reprogramming in NAFLD-induced HCC. The authors have discussed lipid, glucose and amino acid metabolism at length, but all of these have further not been individually and sequentially connected to epigenetics. It would be better if the authors focus on only one type of biomolecule class (lipid, glucose and amino acid metabolism) and provide more comprehensive details about the involvement of epigenetic mechanisms in this particular metabolic reprogramming.

Response 6: We appreciate the reviewer’s insightful suggestion and agree that it would be useful to demonstrate that only one type of biomolecule class is focused on and provide more comprehensive details about how the involvement of epigenetic mechanisms in this metabolic reprogramming; however, such an analysis is beyond the scope of this section, which aims in this section only to present recent studies on metabolic reprogramming in MAFLD-induced HCC. A bidirectional regulatory mechanism between metabolic remodeling and epigenetics exists in tumors. In section 4, "Interactions between epigenetic modifications and HCC cell metabolism," we link epigenetics to metabolism in detail and sequence. Given the greater number of reports on epigenetics mediating the unique metabolic microenvironment of tumors in the field of MAFLD-HCC, we chose to focus on the former. Nevertheless, we have recognized this limitation, so we have extensively reviewed the relevant literature, but unfortunately, the available literature on this subject is quite limited. In our future research, we will conduct a relevant study to understand this issue in greater depth and detail. Thank you very much for your comments.

Comments 7: Figure 3. The figure contains no meaning with regard to the graphical representation and the legend is not well explained. Please modify.

Response 7: We sincerely thank you for your careful reading. We were sorry for our careless mistakes. Thank you for your reminder. As suggested by the reviewer, The legend for Figure 4 (formerly Figure 3) has been added and explained. [Page 22]

Response to Comments on the Quality of English Language

Point 1: Line 49: Change “which is mostly reversible and highly plasticity” to “which is mostly reversible and highly plastic.” The former statement is grammatically inaccurate.

Response 1:  Thanks for your careful checks. We are sorry for our carelessness. In our resubmitted manuscript, the typo is revised. [Page 2, paragraph 1, and line 52]

Point 2: Line 855. Change metabonomics to metabolomics.

Response 2:  We sincerely thank the reviewer for careful reading. As suggested by the reviewer, we have corrected the“metabonomics” into “metabolomics”. [Page 26, paragraph 22, and line 939]

Round 2

Reviewer 2 Report

Comments and Suggestions for Authors

All concerns addressed, ready for acceptance.